



# Exploring the possible role of satellite-based rainfall data to estimate inter- and intra-annual global rainfall erosivity

Nejc Bezak[1], Pasquale Borrelli[2,3], Panos Panagos[4]

[1]University of Ljubljana, Faculty of Civil and Geodetic Engineering, Slovenia
[2]Department of Earth and Environmental Sciences, University of Pavia, Italy
[3]Department of Biological Environment, Kangwon National University, Chuncheon 24341, Republic of Korea
[4]European Commission, Joint Research Centre (JRC), Ispra, Italy

*Correspondence to*: Nejc Bezak (nejc.bezak@fgg.uni-lj.si)

**Abstract.** Despite recent developments in modelling global soil erosion by water, to date no substantial progress has been
made towards more dynamic inter- and intra-annual assessments. In this regard, the main challenge is still represented by the
limited availability of high temporal resolution rainfall data needed to estimate rainstorms rainfall erosivity. As this data
scarcity is likely to characterize the upcoming years, the suitability of alternative approaches to estimate global rainfall
erosivity using satellite-based rainfall data was explored. For this purpose, the high spatial and temporal resolution global
precipitation estimates obtained with the NOAA CDR Climate Prediction Center MORPHing technique (CMORPH) were
used. Alternatively, the erosivity density (ED) concept was used to estimate global rainfall erosivity as well. The obtained
global estimates of rainfall erosivity were validated against the pluviograph data included in the Global Rainfall Erosivity
Database (GloREDa).

Overall, results indicated that the CMORPH estimates have a marked tendency to underestimate rainfall erosivity when
compared to the GloREDa estimates. The most substantial underestimations were observed in areas with the highest rainfall
erosivity values. At continental level, the best agreement between annual CMORPH and interpolated GloREDa rainfall
erosivity map was observed in Europe. Worse agreement was detected for Africa and South America. Further analyses
conducted at monthly scale for Europe revealed seasonal misalignments, with the occurrence of underestimation of the
CMORPH estimates in the summer period and overestimation in the winter period compared to GloREDa. The best agreement
between the two approaches to estimate rainfall erosivity was found for autumn, especially in Central and Eastern Europe.
Conducted analysis suggested that satellite-based approaches for estimation of rainfall erosivity appear to be more suitable for
low-erosivity regions, while in high erosivity regions and seasons (>1,000-2,000 MJ mm ha$^{-1}$ h$^{-1}$ yr$^{-1}$), the agreement with
estimates obtained from pluviograph data such as GloREDa is lower.

Concerning the ED estimates, this second approach to estimate rainfall erosivity yielded better agreement with GloREDa
estimates compared to CMORPH. The application of a simple-linear function correction of the CMORPH data was applied to
provide better fit to the GloREDa and correct systematic underestimation. This correction improved the performance of the
CMORPH but in areas with the highest rainfall erosivity rates the underestimation was still observed. A preliminary trend





analysis of the CMORPH rainfall erosivity estimates was also performed for the 1998-2019 period. According to this trend analysis, increasing and statistically significant trend was more frequently observed than decreasing trend.

**Keywords**: Rainfall erosivity; R-factor; satellite-based precipitation; CMORPH; GloReDa; Erosivity Density

**1 Introduction**

Rainfall erosivity is among the main drivers of soil erosion, which can be characterized by large spatial and temporal variability (Angulo-Martínez and Beguería, 2012; Ballabio et al., 2017; Bezak et al., 2021; Cui et al., 2020; Panagos et al., 2017; Verstraeten et al., 2006). In order to obtain robust rainfall erosivity estimates, high temporal resolution rainfall data is needed (Panagos et al., 2015; Yin et al., 2017). However, according to Panagos et al. (2017) the availability of stations with high-

frequency data that can be used to estimate rainfall erosivity is in average relatively low with many sectors of the world. Therefore, in areas with scarce data availability, remotely measured precipitation data can be very useful in order to estimate rainfall erosivity (Ganasri and Ramesh, 2016; Li et al., 2020). Alternatively, approaches using simpler and less data-demanding methods such as the erosivity density (ED) (Nearing et al., 2017; Panagos et al., 2015, 2016b) can also represent a viable option. Another condition that determines the accuracy of the rainfall erosivity estimates is the low availability of high-temporal-

resolution rainfall data. Ideally, high-frequency (e.g., 1-minute) measurements obtained using optical disdrometers are necessary to quantify the rainfall kinetic energy (Mineo et al., 2019; Nel et al., 2010; Petan et al., 2010; Sanchez-Moreno et al., 2014) of a given storm and to calculate its rainfall erosivity. However, such measuring equipment is not very frequently available in the regional or national measuring networks. Therefore, due to instrumental limitations, rainfall erosivity is communally estimated using hourly or sub-hourly rainfall records (generally ranging from 5 to 60-min) collected by tipping

buckets, or pluviographs, which do not provide information about raindrop size distribution (Panagos et al., 2016a; Petan et al., 2010; Petek et al., 2018). This kind of data is then used together with empirically developed equations that relate rainfall kinetic power and intensity (Brown and Foster, 1987; Carollo et al., 2017; Petan et al., 2010) to obtain rainfall erosivity estimates. Alternatively, rainfall erosivity estimates can also be performed based on the rainfall volume, instead of the intensity, using daily, monthly or annual rainfall data (Renard and Freimund, 1994; Yu and Rosewell, 1996). However, it is worth

mentioning that the accuracy of rainfall erosivity estimates decreases with the increase of the temporal data resolution (i.e., from 1-min to hourly, daily, monthly or annual data). Currently, due to the data scarcity most rainfall erosivity assessments based on long-term estimates including a period of at least 10-years are limited to few regions (Angulo-Martínez and Beguería, 2012; Nearing et al., 2015; Panagos et al., 2015, 2017), leaving large sectors of the world under-researched. In this regard, a step forward is needed to enable the generation of year-by-year and sub-annual rainfall erosivity assessments for under-

researched national or larger scale study areas.

Recent studies have already explored the possibility to estimate rainfall erosivity using satellite-based products at regional (Li et al., 2020) and national scale (Chen et al., 2021; Kim et al., 2020), indicating their sources of uncertainties and a generally limited accuracy (Aghakouchak et al., 2012; Ghajarnia et al., 2018; Prakash, 2019; Prakash et al., 2015; Rahmawati and



Lubczynski, 2018; Seo et al., 2018; Wei et al., 2018). A promising alternative to the often-limited rain-gauge data may be
represented by satellite-based precipitation estimates, which currently have both adequate temporal and spatial resolution
(Chen et al., 2021; Kim et al., 2020; Li et al., 2020; dos Santos Silva et al., 2020; Teng et al., 2017). Moreover, once further
developed and fully operational, the satellite-based methods to estimate rainfall erosivity will have lower purchasing and
processing costs compared to the current ones. In addition, satellite-based rainfall erosivity estimates could be especially useful
in regions where rainfall erosivity estimates are currently very limited such as some sizable sectors of Africa, Asia and South
America.

In this study, the aim was to deepen the research on the use of satellite-based rainfall data to estimate rainfall erosivity
performing a first inter- and intra-annual global scale assessment. The GloREDa data (Panagos et al., 2017) was used to
evaluate both a) the rainfall erosivity estimates obtained by satellite-based rainfall data (i.e. CMORPH) and b) rainfall erosivity
using the ED concept. Finally, a temporal trend analysis of global rainfall erosivity is presented together with corrections
between data based on the CMORPH  and GloREDa databases.

## 2 Data and methods

### 2.1 CMORPH

The CMORPH product is a reprocessed and bias corrected global precipitation dataset covering the area between the 60°S and
60°N parallels, with a 30-min time step and a spatial resolution of 8 km x 8 km (Xie et al., 2021, 2017). The CMORPH data
is developed by the National Oceanic and Atmospheric Administration (NOAA) and covers the period from 1998 onwards. In
general, this method used the precipitation estimates derived from the low Earth orbit satellite-based passive microwave
observations (Kim et al., 2020). Additionally, the geostationary satellite infrared imagery was used to account for possible
coverage issues (Kim et al., 2020). Since CMORPH provides an estimate of the 30-min precipitation, each 30-min rainfall rate
was assumed to be constant during this time interval (Kim et al., 2020; Xie et al., 2021). This dataset has been already applied
for several practical applications such as validation of the climate model simulations, identification of climate extremes, forcing
numerical weather models, characterization of the global precipitation (Xie et al., 2021). Additionally details about the
methodology can be found in existing literature (Chen et al., 2020; Xie et al., 2017, 2021).

### 2.2 GloREDa database

The Global Rainfall Erosivity Database (GloREDa) was created with the objective to develop the first ever global rainfall
erosivity map using high-temporal resolution data (Panagos et al., 2017) and to move towards a new generation of RUSLE-
based soil erosion assessments for present (Borrelli et al., 2017) and future land use dynamics (Borrelli et al., 2020). GloREDa
contains annual rainfall estimates for 3,625 stations from 63 countries with temporal resolution ranging from 1-min to 60-min
(Panagos et al., 2017). The data sample lengths ranged from 5 to 52 years with a mean value of around 17 years with most of
the data covered the period from 2000 to 2010 (Panagos et al., 2017). The number of stations in different continents greatly





varied from around 5% (i.e. South America and Africa) to around 48% (i.e. Europe). Based on the station data and applying
the Gaussian Process Regression model, the global rainfall erosivity map was also prepared (Panagos et al., 2017). Therefore,
in the scope of this study, both the station (i.e. point) estimated annual rainfall erosivity and a global rainfall erosivity map
(Panagos et al., 2017) were used. The spatial resolution of global rainfall erosivity map prepared by Panagos et al. (2017) is
30 arc-seconds (i.e. around 1 km at the Equator). The Rainfall Erosivity Database on the European Scale (REDES) is the
predecessor of GloREDa as it was developed in 2015 (Panagos et al., 2015). As the REDES has made available the monthly
erosivity values, here the monthly rainfall erosivity maps of Europe were also used (Ballabio et al., 2017). All datasets are
available in European Soil Data Centre (ESDAC) (Panagos et al., 2012).

**2.3 Rainfall erosivity calculation**

In order to calculate the annual and monthly rainfall erosivity for each grid cell that is covered by the CMORPH product the
time series with a 30-min time step were extracted from the original CMORPH dataset (Xie et al., 2021). For each grid cell
covered by the CMORPH a 30-min precipitation time series [mm/h] were extracted for the 1998-2019 period. The erosive
events were defined according to the procedure described in the Revised Universal Soil Loss Equation (RUSLE) handbook
(Renard et al., 1997). Thus, two events were separated in case of less than 1.27 mm of rain within 6 h. Only erosive rainfall
events with more than 12.7 mm of rain in total or 6.35 mm in 15 min were considered in the calculations (Kim et al., 2020;
Renard et al., 1997). In order to calculate the specific kinetic energy $e_b$ [MJ ha$^{-1}$ mm$^{-1}$] the Brown and Foster (1987) equation
was applied since this equation was also used by Panagos et al. (2017):

$$e_B = 0.29 \cdot [1 - 0.72 \cdot \exp(-0.05 \cdot I)] , \qquad (1)$$

where $I$ is rainfall intensity [mm h$^{-1}$]. In order calculate the annual rainfall erosivity R-factor [MJ mm ha$^{-1}$ h$^{-1}$ yr$^{-1}$] next two
equations were also used (Renard et al., 1997):

$$E = e_B \cdot I \cdot \Delta t, \qquad (2)$$

$$R = \frac{\sum_n E \cdot I_{30}}{N} , \qquad (3)$$

where $E$ is kinetic energy of individual erosive event [MJ ha$^{-1}$], $\Delta t$ is the time interval [h] and $I_{30}$ is the maximum 30-minute
intensity [mm h$^{-1}$] of erosive event $n$, which occurred within a time span of $N$ years. This procedure was repeated for all grid
cells covered by the CMORPH product.

**2.4 Erosivity density (ED) and ERA5**

The erosivity density (ED) concept was firstly introduced by Kinnell (2010) and was used also in the scope of the enhanced
RUSLE approach named RUSLE2, which led to improvements in rainfall erosivity mapping (Dabney et al., 2012; Nearing et
al., 2017). The ED is defined as ratio between annual or monthly rainfall erosivity and annual or monthly precipitation (Panagos
et al., 2016b). Thus, ED is calculated as the ratio of rainfall erosivity (R) and rainfall depth (P) (Nearing et al., 2017):



$$ED = \frac{R}{P},$$ (4)

Since the introduction of the ED, it has been applied in numerous studies (Diodato et al., 2019; Kinnell, 2019; Nearing et al., 2017; Panagos et al., 2016b). The global rainfall erosivity map obtained by Panagos et al. (2017) was used in this study to obtain a global rainfall erosivity density map (ED). In order to calculate rainfall volume for specific years, the ERA5 reanalysis product was used.

The ERA5 is the one of the latest reanalysis products produced by the European Centre for Medium-Range Weather Forecasts (ECMWF) that provide atmospheric, land-surface and sea-state data. ERA5 includes a large amount of historical observations and it provides long-term solution for ED estimation. The reanalysis data combine model data and observations across the globe into a complete and consistent dataset based on the laws of physics (ERA5, 2021a). Therefore, the ERA5 product is widely used for different purposes (Reder and Rianna, 2021; Sutanto et al., 2020; Tang et al., 2020). The monthly temporal

resolution on single level was used while the horizontal resolution was 0.25° x 0.25°. The temporal coverage used in this study was from 1979 until 2020. Comparison with the CMORPH and GloREDa was made using the 1998-2019 period. Additional information can be found in existing literature (An et al., 2020; ERA5, 2021b; Tang et al., 2020). ERA5 is updated regularly (i.e., monthly updates), which makes it the best option for the dynamic rainfall erosivity assessment at global scale using the ED concept. In case of the ED concept, the annual and monthly ED maps (Ballabio et al., 2017; Panagos et al., 2017) were

multiplied with mean monthly or annual precipitation estimates provided by the ERA5.

## 2.5 Data evaluation

The performance of the rainfall erosivity derived using the CMORPH product and ED concept was evaluated using the GloREDa point dataset (Panagos et al., 2017). This evaluation was performed for the period 1998-2019 and it was performed at global, continental, catchment and local scale. For the later, the values of point data (stations) were compared against values

derived at this location from both methodologies. At catchment scale, the HydroSHEDS catchment boundaries at 3[rd] level were used (Lehner and Grill, 2013). The idea of using 3[rd] level catchment boundaries was to evaluate if the accuracy of the CMORPH and ED derived rainfall erosivity changes with scale (i.e., from global to large regional or even point scale). Moreover, more detailed comparison was made for Europe since monthly rainfall erosivity maps (REDES) are also available (Ballabio et al., 2017) and were used for the comparison as well.

In the data evaluation process the following metrics were calculated: Pearson correlation coefficient, percent bias and Gini coefficient. The Pearson correlation coefficient is a measure of linear correlation between two data sets. The percent bias is a measure of mean tendency of the modelled data to be smaller or larger than observed data. The Gini coefficient is a scalar metrics that can be derived based on the Lorenz curve and is frequently used in economy to describe the inequality of wealth (Gini, 1914; Lorenz, 1905; Masaki et al., 2014). The Gini coefficient ranges from 0 to 1 where a value close 1 and 0 indicates

significant inequality and no inequality, respectively (Masaki et al., 2014). Thus, the idea behind using the Gini coefficient was to use additional metric that describes the distribution of rainfall erosivity in the selected area (e.g., the distribution of





rainfall erosivity grid cells at catchment or continental scale). Therefore, the Gini coefficient can be used as an indicator of the rainfall erosivity spatial patterns. Figure 1 shows an example of different Gini coefficient values for three examples. In the first one, there are similar grid values and Gini coefficient is close to 0, second example with significant inequality where Gini coefficient is close to 1 and the third example with more diverse grid values where Gini coefficient is around 0.5.


Since the spatial resolution of used maps (CMORPH, GloREDa and REDES) were not the same, the GloREDa and REDES maps (and the ED) were resampled to the same grid system extent and resolution that was used by the CMORPH using the mean value (cell area weighted) method that is included in the SAGA GIS software (SAGA GIS, 2021). The same applied for the ERA5 product that was also resampled to the same grid system using B-spline interpolation (SAGA GIS, 2021). Therefore,

the above described comparison at global, continental, regional and point scales was made using the resampled GloREDa and REDES maps (Panagos et al., 2017). A preliminary investigation was done to estimate the effect of the resampling on the mean global rainfall erosivity. The global mean rainfall erosivity using GloREDa map (1 km spatial resolution) was 2,190 MJ mm ha$^{-1}$ h$^{-1}$ yr$^{-1}$ while in case of resampled (i.e., mean) data at 10 km resolution this value equaled to 2,260 MJ mm ha$^{-1}$ h$^{-1}$ yr$^{-1}$. Thus, resampling lead to around 3% difference in the global mean value. However, further aggregation of the GloREDa data

lead to larger differences.

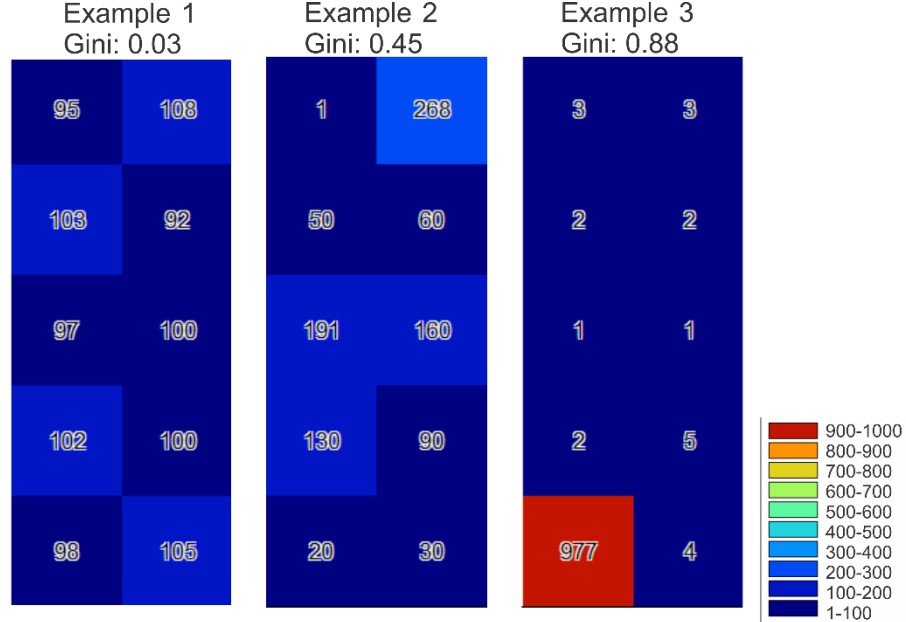

**Figure 1: Gini coefficients for three examples of 10 grid cells with the same mean value (i.e., 100) as an illustration of the added value of the Gini coefficient.**

**2.6 Trends**

Based on the annual CMORPH and ED global rainfall erosivity maps for specific years in the period from 1998-2019, the Mann-Kendall trend test was also calculated for each grid cell. The Mann-Kendall test is one of the most widely applied tests for detection of changes in the environmental data (Burn and Hag Elnur, 2002; Rodrigues da Silva et al., 2016). Detailed





description of the Mann-Kendall test can be found in literature (Burn and Hag Elnur, 2002; Hamed, 2008; McLeod, 2011).

The objective was to identify areas where detected trend in the annual rainfall erosivity data was positive or negative with the

significance level of 0.05.

## 3 Results

### 3.1 Spatial distribution of annual rainfall erosivity

The mean global annual rainfall erosivity using the CMORPH (Figure 2a) data is 1,236 MJ mm ha$^{-1}$ h$^{-1}$ yr$^{-1}$, with a standard

deviation of 1,895 MJ mm ha$^{-1}$ h$^{-1}$yr$^{-1}$. The mean global annual rainfall erosivity using the erosivity density (ED) approach

(Figure 2b) is 2,480 MJ mm ha$^{-1}$ h$^{-1}$ yr$^{-1}$. As can be inferred from Figure 2, and further indicated in Table 1, at continental

level the highest values of rainfall erosivity were estimated for North America, while the smallest ones were estimated for

Europe in the case of both R and ED approaches.

Concerning inequality, the largest value of rainfall erosivity inequality obtained using the Gini coefficient was detected for

Asia, followed by Africa and Oceania, while the smallest value was observed for Europe (Table 1).


**Table 1: Mean, standard deviation and Gini coefficient of the global rainfall erosivity maps derived using the CMORPH and ED.**

| Continent | CMORPH | | | ED | | |
|---|---|---|---|---|---|---|
| | Mean [MJ mm ha$^{-1}$ h$^{-1}$ yr$^{-1}$] | St. dev. [MJ mm ha$^{-1}$ h$^{-1}$ yr$^{-1}$] | Gini [/] | Mean [MJ mm ha$^{-1}$ h$^{-1}$ yr$^{-1}$] | St. dev. [MJ mm ha$^{-1}$ h$^{-1}$ yr$^{-1}$] | Gini [/] |
| Africa | 1,038 | 1,619 | 0.62 | 3,037 | 3,277 | 0.56 |
| Asia | 1,138 | 2,242 | 0.73 | 2,255 | 3,554 | 0.69 |
| Oceania | 1,004 | 1,207 | 0.48 | 1,630 | 1,916 | 0.51 |
| Europe | 614 | 490 | 0.36 | 646 | 500 | 0.36 |
| North America | 892 | 1,072 | 0.53 | 1,748 | 1,947 | 0.52 |
| South America | 2,556 | 2,179 | 0.41 | 6,640 | 3,961 | 0.32 |



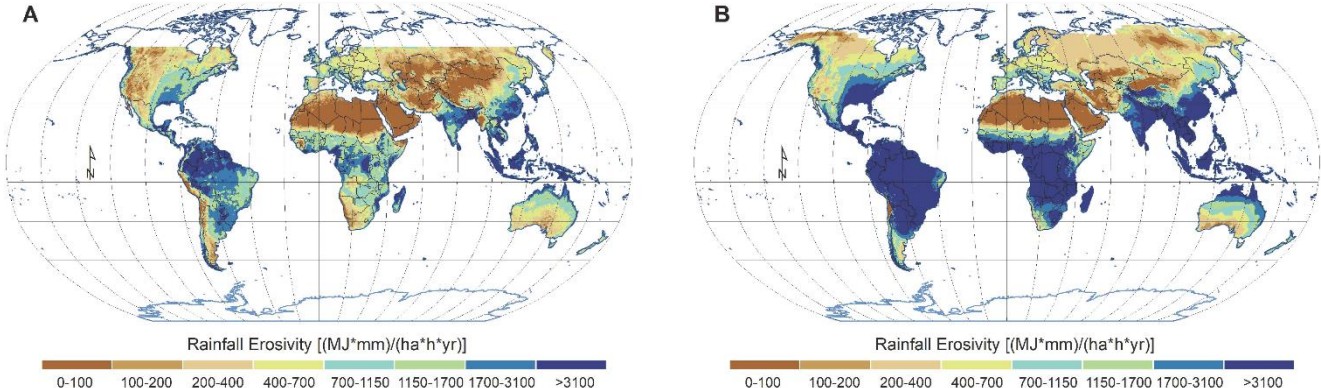

**Figure 2: Mean global rainfall erosivity map for the 1998-2019 period based on the CMORPH product (a) and ED concept using the ERA5 (b).**

**3.2 Temporal trends in rainfall erosivity**

Table 2 shows mean and standard deviation of monthly rainfall erosivity derived using the CMORPH product. One can notice that the highest rainfall erosivity values were obtained in July followed by August and the lowest in November (Table 2). Although the monthly rainfall erosivity is relatively uniformly distributed through the year.

The temporal trends for both CMORPH and ED derived rainfall erosivity datasets were also calculated. The annual rainfall

erosivity in the period 1998-2019 ranged from 990 to 1,440 MJ mm ha$^{-1}$ h$^{-1}$ yr$^{-1}$ using the CMORPH data (Figure 3; Figure S1). Using the ED concept for global rainfall erosivity, the mean value ranged from 2,380 to 2,646 MJ mm ha$^{-1}$ yr$^{-1}$ (Figure 3; Figure S1) for the period 1981-2019, while for the period 1998-2019 this variation was a slightly smaller (2,380 to 2,602 MJ mm ha$^{-1}$ h$^{-1}$ yr$^{-1}$). In addition, the fluctuation of the mean annual erosivity in case of the ED concept were smaller compared to the CMORPH (Figure 3), condition which can be related to the fact that the adopted ED concept used a constant ED map

for the entire period, while only annual rainfall (i.e. ERA5) changed from year to year.

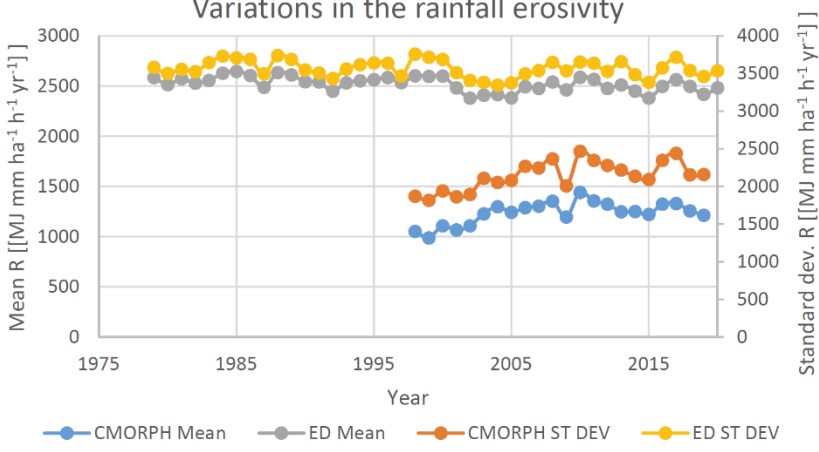

**Figure 3: Trend analysis for the mean and standard deviation (ST DEV) for annual rainfall erosivity (R).**





**Table 2: Global monthly rainfall erosivity values using the CMORPH product, mean and standard deviation are shown.**

| Month | Mean [MJ mm ha$^{-1}$ h$^{-1}$ mo$^{-1}$] | St. dev. [MJ mm ha$^{-1}$ h$^{-1}$ mo$^{-1}$] |
|---|---|---|
| January | 101 | 213 |
| February | 96 | 198 |
| March | 99 | 200 |
| April | 96 | 193 |
| May | 99 | 208 |
| June | 109 | 227 |
| July | 124 | 245 |
| August | 120 | 228 |
| September | 102 | 204 |
| October | 97 | 195 |
| November | 93 | 200 |
| December | 100 | 221 |

**3.3 Data evaluation**

**3.3.1 Comparison at global scale**

For most continents, relatively large differences in the mean long-term annual rainfall erosivity between GloREDa and CMORPH were observed, while smaller differences were observed between ED and GloREDa (Table 3). The largest differences in case of the CMORPH were detected for South America, Africa and North America (Table 3). In case of the ED concept, the largest differences were calculated for Asia and South America. On the other hand, much better agreement between the CMORPH and GloREDa maps was observed for Europe and partly for Asia (Table 3). In terms of Gini coefficient, better agreement between the CMORPH and GloREDa and ED and GloREDa can be seen compared to the mean annual rainfall erosivity (Table 3). Thus, it seems that distribution of the erosivity in case of the CMORPH, ED concept and GloREDa was relatively (i.e. smaller bias) similar despite the fact that the GloREDa map was based on the interpolation. It should be noted that the ED provided better fit to the GloREDa compared to the CMORPH at all continents while the mean values can be quite different, especially in case of the CMORPH (Table 3).





**Table 3: A comparison between the CMORPH, ED concept and GloREDa derived global rainfall erosivity at continental scale.**

| Continent | CMORPH bias compared to GloREDa [%] | | | ED bias compared to GloREDa [%] | | | GloREDa [MJ mm ha$^{-1}$ h$^{-1}$ yr$^{-1}$] | | |
|---|---|---|---|---|---|---|---|---|---|
| | Mean | St. dev. | Gini [/] | Mean | St. dev. | Gini [/] | Mean | St. dev. | Gini [/] |
| Africa | -66 | -46 | 17 | -1 | +9 | 6 | 3,055 | 2,992 | 0.53 |
| Asia | -38 | -23 | 4 | +23 | +22 | -1 | 1,839 | 2,925 | 0.70 |
| Australia-Oceania | -40% | -39 | -8 | -3 | -3 | -2 | 1,676 | 1,975 | 0.52 |
| Europe | +11% | +17 | 0 | +17 | +20 | 0 | 553 | 418 | 0.36 |
| North America | -47 | -48 | -7 | +4 | -7 | -9 | 1,683 | 2,082 | 0.57 |
| South America | -56% | -36 | 24 | +13 | +17 | -3 | 5,866 | 3,381 | 0.33 |

### 3.3.2 Comparison at regional scale

The HydroSHEDS catchment boundaries at the 3$^{rd}$ level were used to compare the data of CMORPH with the GloREDa at regional scale. Thus, the global land surface was divided into 288 sub-catchments at the 3$^{rd}$ level with mean catchment area of

around 460,000 km$^2$. Thus, this can be regarded as very large regional-scale investigation. The results demonstrated that the Pearson correlation between the mean annual rainfall erosivity at the sub-catchment level between the CMORPH and GloREDa was 0.81 ($R^2 = 0.66$ with p-value < 0.01). Moreover, the mean bias was around -50% in case that GloREDa data was considered as observed data. In case of Gini coefficient, the Pearson correlation coefficient was 0.56 ($R^2 = 0.31$ with p-value < 0.01), while the mean bias equaled to 45%. Therefore, the CMORPH yielded more unequal (i.e., larger Gini coefficient) spatial

erosivity patterns compared to the GloREDa, which was based on the interpolation.

The comparison between ED concept and GloREDa revealed that Pearson correlation coefficient equaled to 0.95 ($R^2 = 0.90$ with p-value < 0.01) and the mean bias was 7%. In terms of Gini coefficient, the Pearson correlation coefficient and the mean percent bias were 0.91 ($R^2 = 0.83$ with p-value < 0.01) and 3.4%, respectively. Therefore, GloREDa and ED maps have similar spatial erosivity patterns. Furthermore, two examples of good (Figures 4) and bad (Figures 5) agreement between the

CMORPH, ED and GloREDa are presented.





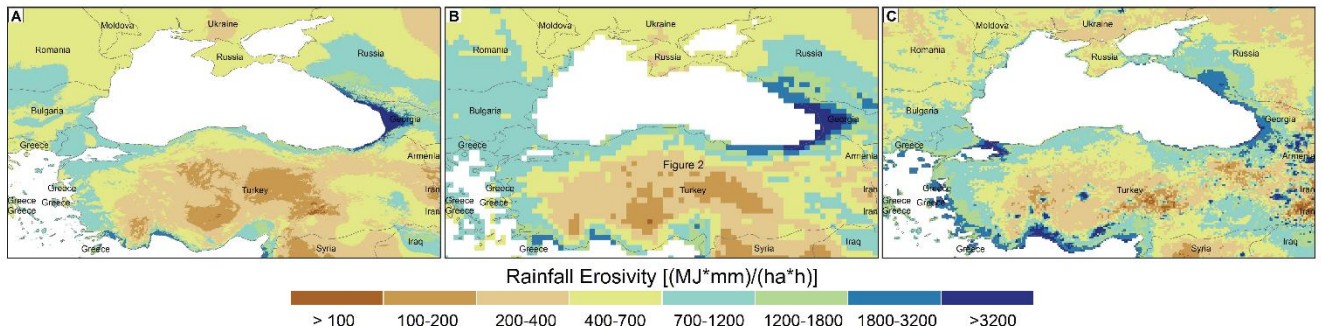

**Figure 4: An example of relatively good agreement between the GloREDa, ED and CMORPH maps for parts of Eastern Europe and Turkey.**

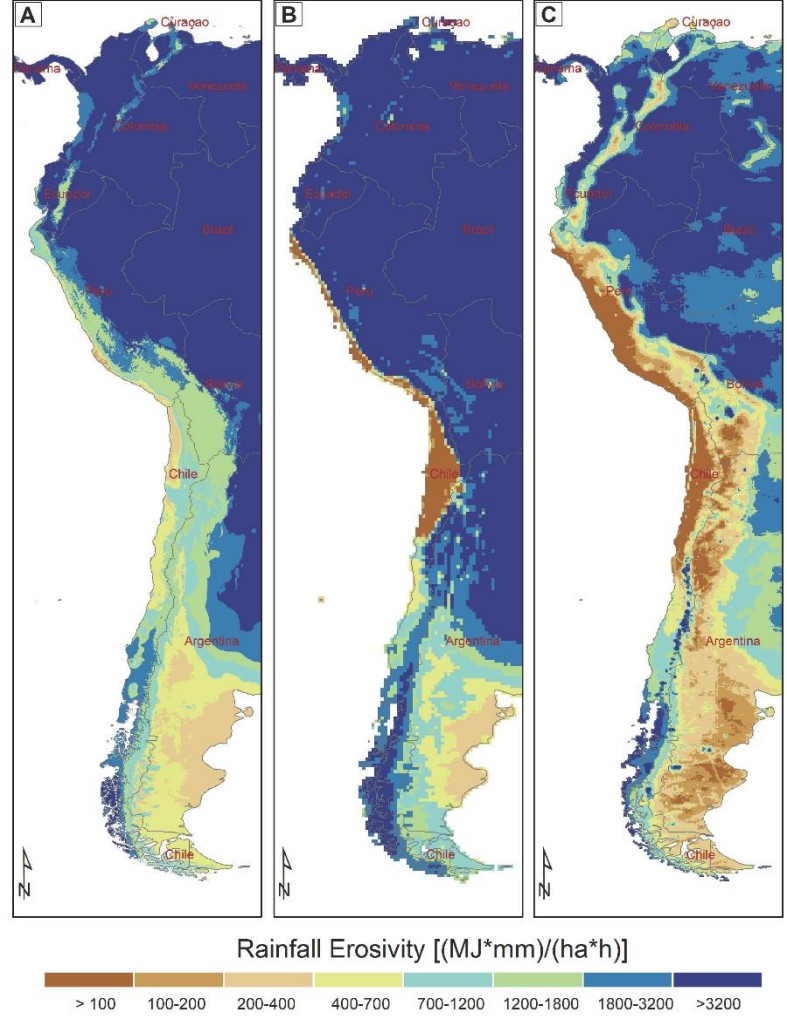

**Figure 5: An example of worse agreement between the GloREDa, ED and CMORPH maps for the parts of Southern America.**





In Europe, the best agreement between the CMORPH and GloREDa was found (Table 3), the smallest uncertainty in GloREDa and the highest station density plus and available monthly rainfall erosivity estimates (Ballabio et al., 2017). For those reasons, a more in-depth assessment was made for Europe. According to GloREDa, the mean annual rainfall erosivity in Europe (i.e.,

without Russia) was 668 MJ mm ha$^{-1}$ h$^{-1}$ yr$^{-1}$ with a standard deviation of 429 MJ mm ha$^{-1}$ h$^{-1}$ yr$^{-1}$ (Figure 6). According to the CMORPH product, the mean and standard deviation equaled to 752 and 533 MJ mm ha$^{-1}$ h$^{-1}$ yr$^{-1}$, respectively (Figure 6). Additionally, the ED concept yielded a mean annual rainfall erosivity value of 804 with a standard deviation of 541 MJ mm ha$^{-1}$ h$^{-1}$ yr$^{-1}$ (Figure 6). Moreover, calculated Gini coefficient using all grid cells was 0.31 in all cases (Figure 6). Thus, it can be seen that the CMORPH product yielded relatively similar erosivity distribution across Europe compared to the GloREDa,

which means that all maps have similar level of inequality (i.e., non-uniform distribution of rainfall erosivity). A bit larger rainfall erosivity values were obtained using the ED concept. It should be noted that part of these differences can be attributed to the fact that GloREDa dataset mostly used data in the 2000-2010 period (Panagos et al., 2017). Moreover, in some areas (e.g., Italy, Balkan Peninsula, parts of Eastern Europe) spatial patterns in all cases were similar, although CMORPH product and ED concept yielded slightly variable rates (Figure 6). On the other hand, the CMORPH yielded higher annual rainfall

erosivity values compared to the GloREDa map in case of parts of British Isles and parts of Eastern Europe (Figure 6). Furthermore, CMORPH tends to underestimate areas with relatively high rainfall erosivity such as Alpine region, Spain, Italy, and other parts of the Mediterranean basin (Figure 6).

As in the EU there are available monthly erosivity maps, the comparison between the CMORPH and ED maps was conducted (Table 4). A better agreement CMORPH and GloREDa for autumn compared to summer and winter was found (Table 4). The

ED concept yielded higher rainfall erosivity values in almost all months, which also resulted in higher differences at annual level (Table 4). This could be attributed to the underestimation of the WorldClim V1 map (Beck et al., 2020). In addition, GloREDa has lower values compared to REDES in Europe. However, there is a stronger linear dependence between monthly ED and GloREDa than the one between CMORPH and GloREDa (Table 4).

Moreover, Figure 7 shows monthly erosivity values for selected months where three cases were selected (i.e. under-, over-

estimation and almost complete agreement between CMORPH and GloREDa).In July, the CMORPH product in the Alpine region generally yielded smaller erosivity values compared to both the monthly erosivity maps prepared by Ballabio et al. (2017) and the ED map (Figure 7). The same conclusion is reached for other regions such as parts of Western Europe or Iberian Peninsula (Figure 7). In December, parts of Eastern Europe have better agreement among three maps (Figure 7). On the other hand, October is the month with the best agreement among the three tested maps (Figure 7). For October, the best agreement

is found in parts of Eastern and Central Europe while the worst was detected in parts of Iberian Peninsula in October (Figure 7). The CMORPH derived rainfall erosivity is in some cases more equally distributed (i.e. winter) and in other cases more unequally distributed (i.e. summer) compared to the GloREDa while in case of the ED concept and GloREDa the derived Gini coefficients are similar throughout the year (Table 4).





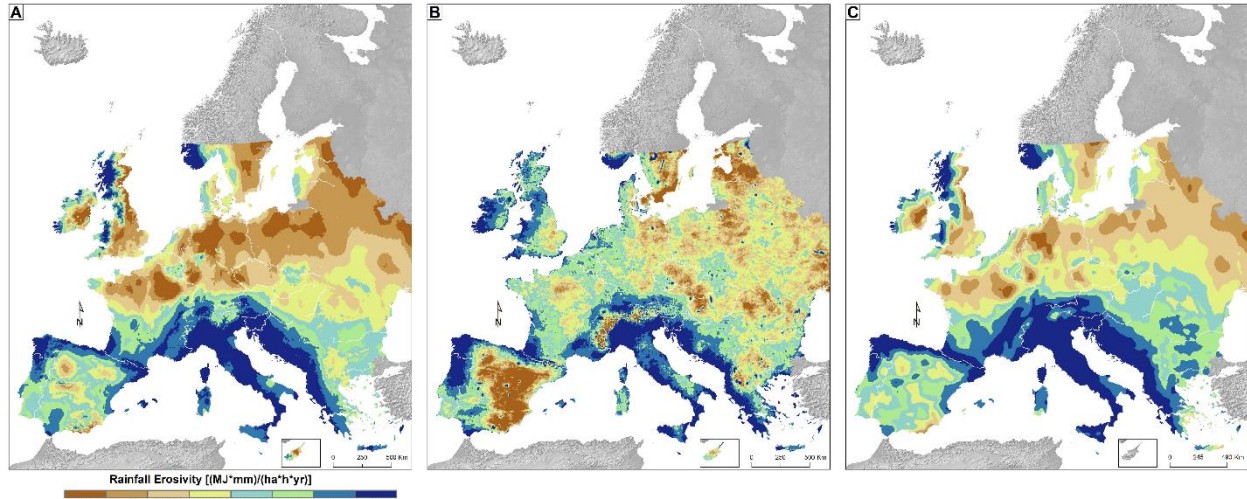

**Figure 6: Comparison between the GloREDa rainfall erosivity map prepared by Panagos et al. (2017) (a), CMORPH derived rainfall erosivity map (b) and ED concept derived map (c) for Europe. Dotted lines indicate longitude and latitude lines separated by 10 degrees.**

**Table 4: Comparison between monthly rainfall erosivity characteristics using Ballabio et al. (2017), ED concept and CMOPRH product. Values in brackets indicate percent bias compared to the GloREDa map.**

| Month | CMORPH bias compared to GloREDa [%] | | | ED bias compared to GloREDa [%] | | | Monthly R (Ballabio et al., 2017) | | |
|---|---|---|---|---|---|---|---|---|---|
| | Mean | St. dev. | Gini | Mean | St. dev. | Gini | Mean [MJ mm ha$^{-1}$ h$^{-1}$ mo$^{-1}$] | St. dev. [MJ mm ha$^{-1}$ h$^{-1}$ mo$^{-1}$] | Gini [/] |
| January | +196 | +164 | -11 | +4 | +3 | -5 | 26 | 36 | 0.63 |
| February | +117 | +105 | -8 | +4 | 0 | -2 | 24 | 37 | 0.65 |
| March | +93 | +74 | -20 | +33 | +49 | +3 | 27 | 43 | 0.64 |
| April | +56 | +71 | -16 | +25 | +26 | +2 | 32 | 34 | 0.51 |
| May | -15 | +25 | +9 | +24 | +43 | +6 | 67 | 40 | 0.32 |
| June | -40 | -29 | +6 | +12 | +20 | +3 | 101 | 66 | 0.35 |
| July | -42 | -25 | +22 | +24 | +38 | +6 | 121 | 72 | 0.32 |
| August | -41 | -35 | +12 | +16 | +25 | +3 | 112 | 72 | 0.33 |
| September | -29 | -29 | -16 | +18 | +36 | +9 | 82 | 80 | 0.44 |
| October | -3 | -21 | -24 | +13 | +20 | 0 | 79 | 90 | 0.54 |
| November | +64 | +46 | -18 | +16 | +38 | +5 | 56 | 74 | 0.61 |
| December | +80 | +23 | -25 | -5 | -7 | -3 | 44 | 70 | 0.67 |




**Figure 7: Comparison between monthly erosivity maps prepared by Ballabio et al. (2017), ED concept maps and CMORPH derived maps for July, October and December. Dotted lines indicate longitude and latitude lines separated by 10 degrees.**





### 3.3.3 Comparison at local scale using GloREDa stations

The station data of GloREDa was also compared with grid cell values at the same location from the derived CMORPH and ED rainfall erosivity datasets. The Pearson correlation coefficient between CMORPH and GloREDa datasets equaled to 0.74 ($R^2 = 0.55$ with p-value < 0.01) and the mean bias was equal to -32% (Figure 8). In general, CMORPH product yielded smaller rainfall erosivity estimates, especially for locations where annual rainfall erosivity exceeded 5,000 MJ mm ha$^{-1}$ h$^{-1}$ yr$^{-1}$ (Figure 8). Additionally, CMORPH product tended to overestimate rainfall erosivity in locations near water bodies, which are also the

points located under the orange line shown in Figure 8. A comparison between the ED concept and the GloREDa yielded a Pearson correlation coefficient of 0.77 ($R^2 = 0.59$ with p-value < 0.01) and the mean bias of 10%. Thus, similarly as in case of global, continental or large-catchment scale, a better agreement between ED concept and GloREDa was detected if compared with CMORPH versus GloREDa.

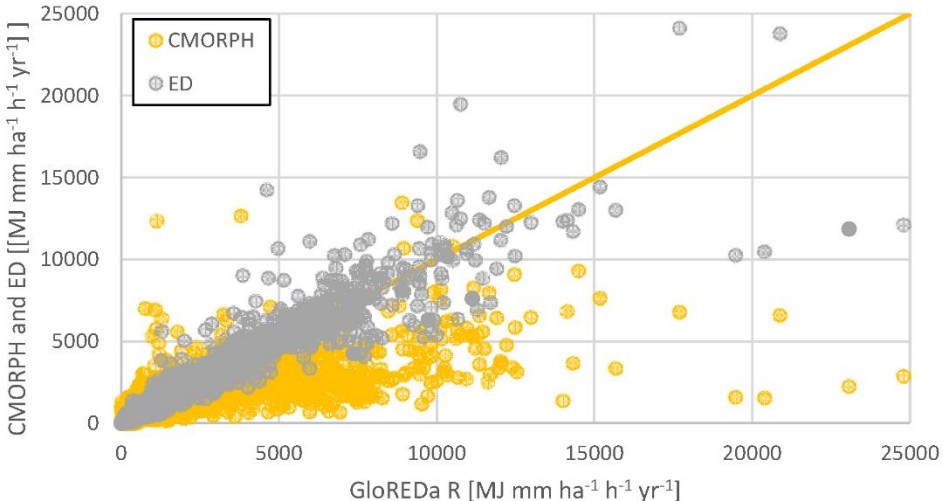

**Figure 8: Comparison between local (i.e. grid cell values) long-term annual rainfall erosivity and stations' values of GloREDa.**

### 3.4 CMORPH data correction using GloREDa

Taking into account the results and comparisons presented above, the attempt to adjust the CMORH rainfall erosivity estimates using the estimates of the GloREDa ground stations database was made. A similar attempt was also already made by Kim et al. (2020) and Wang et al. (2020). In the scope of this study, the corrections were made at global scale using GloREDa stations

with the objective to move CMORPH-GloREDa points as close to the 1:1 line (Figure 9). The best-fitted linear function that was found:

$$CMORPH_{COR} = 0.5*GloREDa \qquad\qquad\qquad (5)$$





After the correction, a much better linear dependence between the corrected CMORPH and GloREDa at station level was observed (Figure 9). The same correction was also applied to the global rainfall erosivity map derived using the CMORPH product (Figure S2) and yielded a global mean rainfall erosivity of 2,490 MJ mm ha$^{-1}$ h$^{-1}$ yr$^{-1}$ with a standard deviation of 3,140 MJ mm ha$^{-1}$ h$^{-1}$ yr$^{-1}$. Even if one notices a much better agreement between the CMORPH$_{COR}$ and GloREDa after the correction, this is relevant only for long-term mean rainfall erosivity assessments. In case of dynamic rainfall erosivity maps (i.e., for specific years or months), different corrections factors should be applied based on the relationship between CMORPH and stations rainfall erosivity for specific years.

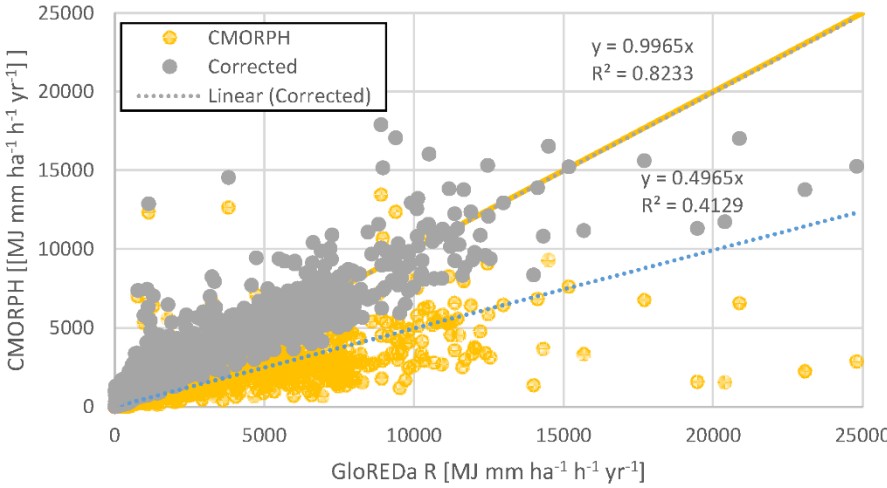


**Figure 9: Comparison between CMORPH and GloREDa datasets at station scale and corrected CMORPH data (CMORPH$_{COR}$).**

**3.5 Temporal global erosivity trends for the period 1998-2019**

The Mann-Kendall test was applied to identify areas with statistically significant (i.e., with the 0.05 significance level) changes (Figure S3) during the period 1998-2019. According to CMORPH erosivity output, 15% of the globe has a statistically significant change (Figure S3). In case of change, most of the regions show a positive trend than negative according to the CMORPH product (Figure S3). Therefore, the positive trend covers 80% of the area where statistically significant change was detected (i.e., around 12% of the total area) (Figure S3) while the rest 20% (3% of the total area) has a negative trend.

On the other hand, the ED concept yielded different results as around 13% of the total area shows a statistically significant trend with positive and negative trends having similar shares (i.e., around 6.5% each). Consistent trends for using both methods (i.e., CMORPH and ED concept) were estimated in parts of North America and Asia while opposite trends were found in parts of Africa, Asia and Europe. A direct comparison with study conducted by Bezak et al. (2020) that investigated rainfall erosivity trends in Europe (1961-2018) was not possible since the investigated periods did not overlap. However, both studies indicated that rainfall erosivity could be increasing (according to the CMORPH product) over the last two decades.





## 4 Discussion

The global mean annual rainfall erosivity as derived by Panagos et al. (2017) totals about 2,190 MJ mm ha$^{-1}$ h$^{-1}$ yr$^{-1}$ (i.e. initial 1-km cell size map), with a standard deviation of 2,974 MJ mm ha$^{-1}$ h$^{-1}$ yr$^{-1}$. The area covered by CMORPH is slightly smaller (between 60°S and 60°N parallels) than the one covered by the GloREDa map (between 60°N and ~75°N parallels, including some parts of Scandinavia, Siberia, Canada). However, a GloREDa global mean rainfall erosivity value about 1.8 higher than the one derived based on the CMORPH data (shown in section 3.1) cannot be explained by the slight difference between the

two study areas. It should be noted that also Kim et al. (2020) reported that the CMORPH derived rainfall erosivity was 1.65 higher than the GloREDa estimates (Panagos et al., 2017) for the United States of America (USA), with some USA regions showing a bias larger than 80% (Kim et al., 2020). More specifically, Kim et al. (2020) reported a mean annual value of 1,260 MJ mm ha$^{-1}$ yr$^{-1}$ for the USA while in this study a mean value of 1,173 MJ mm ha$^{-1}$ h$^{-1}$ yr$^{-1}$ was calculated using slightly different methodology (e.g., different $e_b$-$I$ equation was applied) and different time period. As the station density was quite

low in case of Panagos et al. (2017) study for Africa and North America, this can partly explain the larger differences between CMORH and GloREDa. On the other hand, the largest number of stations was positioned in Europe and also Asia (Panagos et al., 2017) where the agreement between CMORPH and GloREDa was the best (Table 3).

In line with what already discussed by Kim et al. (2020), the insights gained by conducted global analysis suggest that the CMORPH satellite-based rainfall erosivity estimates provide more seamless erosivity distribution without employing

interpolation, uniform and good spatial coverage, 30-time temporal resolution. However, it is also clear that this product has important disadvantages, i.e., overestimated precipitation over water bodies, detection accuracy in hilly terrains can be problematic, accuracy of annual precipitation can be low, relatively bias at event scale can be large.

As noted several studies have indicated that difference between satellite-based products and ground-based precipitation data can be quite significant (Habib et al., 2012; Haile et al., 2015; Jiang et al., 2018). There were numerous studies published that

investigated accuracy of the CMORPH product in terms precipitation. For example, Islam et al. (2020) showed that CMORPH overestimated daily precipitation amount in Australia. Similar conclusion was made by An et al. (2020) for the Yellow River in China or by Wei et al. (2018) for the mainland China. Some studies also showed a significant underestimation of the CMORPH in winter seasons (Gebregiorgis and Hossain, 2015).

Additionally, Palharini et al. (2020) showed that satellite-based products tended to underestimate extreme precipitation, which

has an important effect on the rainfall erosivity. Underestimation of extreme rainfall events was also reported in multiple other studies (Jiang et al., 2019; Rahmawati and Lubczynski, 2018; Stampoulis et al., 2013; Sunilkumar et al., 2015; Wei et al., 2018b). Moreover, Tian et al. (2009) also showed that overestimation was seen for summer (i.e. overestimation of heavy precipitation with intensity over 40 mm/day) and underestimation for winter (i.e. miss of significant amount of light precipitation with intensity lower than 10 mm/day) for the USA. Tian et al. (2009) also found that that hit bias and missed

precipitation were the two dominant error sources. Similar conclusion was also made by Jiang et al. (2018) who pointed the limited detection accuracy  of summer thunderstorms by the CMOPRH product in Shanghai region. On the other hand, the





only study that, to the best of authors knowledge, investigated this satellite-based derived rainfall erosivity (Kim et al., 2020) showed that CMORPH underestimated rainfall erosivity in the USA compared to the GloREDa map. Thus, it is clear that underestimation of the most extreme rainfall events can lead to large differences in the derived rainfall erosivity map. Such

characteristics can also lead to relatively large differences in case that satellite products are used for the flood investigations (Dis et al., 2018). Underestimation of the precipitation amount by the CMOPRH product in the Southern Europe as shown in the Results section was also indicated by some other studies (Skok et al., 2016). Furthermore, also Stampoulis and Anagnostou (2012) pointed out that satellite-based precipitation products accuracy tended to be lower over the mountainous regions such as Alpine region (or Andes). This was especially evident during the cold season (Stampoulis and Anagnostou, 2012) and was

also highlighted in some other studies (Kidd et al., 2012). Also other studies pointed a  detection problem for winter precipitation and high-intensity rainfall events in some parts of Europe (Stampoulis and Anagnostou, 2012). Similarly, also Kidd et al. (2012).

Different examples of good and bad agreement among presented rainfall erosivity maps can be seen around the globe (Figure 4 and Figure 5). Comparing the three maps (i.e., GloREDa, CMORPH and ED) for parts of the Eastern Europe and Turkey

(Figure 4), a relatively good agreement between all the three maps was detected. The main reasons for this good agreement are a) the relatively large number of stations with measured R-factor which contributed to  GloREDa map in countries such as Romania or Turkey (Panagos et al., 2017); b) the relatively flat terrain without major mountainous regions in parts of Eastern Europe and c) the relatively low-medium erosivity ($< 1,000$ MJ mm ha$^{-1}$ h$^{-1}$ yr$^{-1}$). On the contrast, there are many regions where differences are much larger. An example is the Andes mountain region (Figure 5) where the GloREDa includes only 15 stations

in central part of Chile (Panagos et al., 2017) and also gridded precipitation products such as WorldClim underestimate precipitation (Beck et al., 2020).

The ED (based on the ERA5 data) rainfall erosivity estimates showed a better agreement with the GloREDa point estimates. The largest differences between the ED and the GloREDa estimates were observed in Asia, Europe and South America because of the precipitation underestimation in mountainous regions such as Andes, Himalayas and Alps (Beck et al., 2020). The

deviation of  ED compared to GloREDa map could be explained by two main reasons: a) the difference in the spatial resolution of the GloREDa and CMORPH maps as aggregating the 1-km GloREDa map to the 0.25° that is used by the ERA5 yielded to a global mean value of 2,329 MJ mm ha$^{-1}$ h$^{-1}$ yr$^{-1}$; b)  the WorldClim V1 map that was used as input to produce the GloREDa map underestimates the precipitation and that updated version of the WorldClim map (i.e. V2) yields around 10% higher annual global precipitation (Beck et al., 2020).

ED has the following advantages compared to CMORPH approach: a) the ED concept can be used to prepare dynamic rainfall erosivity maps and b) there are no issues with the accuracy near water bodies. On the other hand, ED also has some shortcomings: a) most of the gridded precipitation datasets underestimate precipitation over mountain regions (Beck et al., 2020); b) considers the erosivity/precipitation relationship as constant.


## 5 Study limitations and other products

The density of stations used to produce the GloREDa map is locally low, especially in the African and South American continents. Obviously, this could have an important effect on the produced global rainfall erosivity map (Panagos et al., 2017), and consequently on the results presented in this study since the GloREDa map was here used as a reference. However, to the best of authors knowledge GloREDa is the only global assessment using hourly and sub-hourly rainfall data and the best performing among the global assessments currently available (Panagos et al., 2017). This is due to coarses time step of other

potential global rainfall erosivity sources (Liu et al., 2020). Since the ED concept directly uses the GloREDa map, the results produced by the ED method are directly influenced by the potential shortcomings of the GloREDa and this should be taken into account when making further applications using the ED.

On the other hand, the satellite-based precipitation products have their own sources of uncertainty as highlighted in the previous sections and consequently CMORH significantly underestimates global rainfall erosivity rates. It should be noted that there

are other potential products that could have been used to produce global rainfall erosivity maps and that could yield better results than the CMORPH. For example, MSWEP uses gauge, reanalysis and satellite data sources and it was shown that outperforms some other products such as CMORPH (Beck et al., 2019a, 2019b). Moreover, the Tropical Rainfall Measuring Mission (TRMM) rainfall products can also be used to derive the rainfall erosivity (Li et al., 2020). However, it should be noted that the temporal resolution of these two products is 3 hours, which requires a non-standard RUSLE approach to derive

the rainfall erosivity (Renard et al., 1997). Thus, alternative approaches for the rainfall erosivity estimation are needed. For example, Li et al. (2020) used a modified Brown and Foster equation to calculate the specific kinetic energy and consequently the rainfall kinetic energy. However, this equation was developed based on the case study from China and can be therefore regarded as local (not global) equation. Thus, applying this equation to the global scale could introduce additional uncertainty to the results. Furthermore, one could also apply the correction (i.e. conversion) factor that was suggested by Panagos et al.

(2016b). However, a relatively high value is obtained for the 3 hours duration (i.e. a value of 6.6) and the equation used to calculate the conversion factors was only developed for durations up to 1 hour. Therefore, the results applying the correction factors developed by Panagos et al. (2016b) could lead to the uncertain results. Thus, there is no globally acceptable method for the calculation of the global rainfall erosivity using 3 hours data set. Moreover, these two products also have more coarse spatial resolution compared to the CMORPH, which also has an effect on the detection of the most extreme rainfall events.

There are other potential sources (e.g., reanalysis, satellite-based or combined) with different temporal and spatial resolution that could be additionally tested (Beck et al., 2019a).

## 6 Conclusions

The global rainfall erosivity was assessed using the CMORPH product and the erosivity density (ED) concept. The comparison of the derived maps was performed at global and multiple scales using annual and monthly rainfall erosivity values.





The CMORPH product leads to a marked underestimation of annual rainfall erosivity across the globe, with an average value of 1.8 times lower than the global mean obtained interpolating point information (GloREDa map). The agreement between CMORPH and GloREDa estimates varied significantly among continents. While the best agreement was detected for Europe (i.e., percent bias around 10%) that, on average, has relatively low erosivity values, a considerably lower performance was observed for Africa and South America (i.e., percent bias around -60%). These regions, besides having a higher average rainfall erosivity value than Europe, are also suffering a considerably lower number of measurement stations in the GloREDa database. Interpretation of the results obtained, suggested that satellite-based products such as CMORPH cannot correctly capture the most extreme rainfall events that contribute to the largest proportion of the annual rainfall erosivity in some parts of the globe. Better agreement was generally detected between the ED concept and GloREDa (i.e. percent bias up to around 20%), which can be regarded as an expected result since the ED concept indirectly uses the information from the GloREDa.

A more detailed comparison was performed for Europe, where an investigation was also performed at monthly time scale. Some spatial erosivity patterns were well detected by the CMORPH product in some regions and monthly erosivity values in spring in autumn were relatively close to the ones reported by the monthly erosivity maps prepared by Ballabio et al. (2017) (e.g., in parts of Eastern and Central Europe). Additionally, underestimation and overestimation were detected in summer (percent bias up to -40%) and winter (percent bias up to 100%) compared to the GloREDa, respectively. On the other hand, the ED concept consistently slightly overestimated the GloREDa but yielded better agreement with the GloREDa both temporally and spatially than the CMORPH (i.e. percent bias was in the range of around 30%).

A preliminary trend investigation revealed that around 15% of the investigated area was characterized by the statistically significant change in the annual rainfall erosivity while in around 80% this change was positive (i.e. 12% of the total area) according to the CMORPH product for the 1998-2019 period. According to the ED concept, 13% of the area was characterized by statistically significant trend. In some regions (e.g., parts of South or North America) detected trends were consistent while in other trends were not consistent (e.g., parts of Africa or Asia). Thus, detected trends according to the CMORPH could indicate that rainfall erosivity around the globe has been increasing in the recent decades. However, more detailed investigation is needed in order to confirm or reject this preliminary result.

It should be noted that in case that CMORPH product is used for the preparation of the rainfall erosivity map that would be further used for the soil erosion modelling an uncertainty assessment should be included in such investigation, similarly as in some other scientific disciplines (e.g., Kim et al., 2016; Sun et al., 2018).

Despite the mentioned shortcomings and strong underestimation of the rainfall erosivity in some parts of the globe, the satellite-based precipitation products tend be an interesting option for the estimation of the rainfall erosivity, especially in regions with limited ground data. However, in some regions and in some seasons such products require additional correction to remove bias, which is of course related to the availability of ground-based precipitation. Thus, it is clear that such ground-based high-frequency precipitation measurements are (still) essential for accurate rainfall erosivity estimates, however, one can expect that technological development in the next decades will lead to improved accuracy (Tang et al., 2020) of satellite-based products such as CMORPH. This kind of products could be used as an input to the dynamic soil erosion, which could be used





by relevant stakeholders. At the moment, alternative approaches such as the ED concept can provide more accurate rainfall
erosivity estimates and can be more easily computed as well.

## Funding

N.Bezak would like to acknowledge support of the Slovenian Research Agency (ARRS) through grant P2-0180 and support
from the UNESCO Chair on Water-related Disaster Risk Reduction. Pasquale Borrelli is funded by the EcoSSSoil Project,
Korea Environmental Industry & Technology Institute (KEITI), Korea (Grant No. 2019002820004).

## Acknowledgments

We thank the National Oceanic and Atmospheric Administration (NOAA) for providing the global bias-corrected CMORPH
CDR rainfall data and the European Centre for Medium-Range Weather Forecasts (ECMWF) who have made available the
ERA5 dataset.

## Code and data availability

The CMORPH data can be downloaded at: https://www.ncei.noaa.gov/data/cmorph-high-resolution-global-precipitation-
estimates/access/. ERA5 was downloaded through Copernicus CDS. Rainfall erosivity products were derived from:
https://esdac.jrc.ec.europa.eu/resource-type/datasets. R code can be obtained upon request from the corresponding author.

## Author contributions

All authors developed the concepts of the manuscript, N.B. conducted calculations and wrote the first draft. P.B. and P.P.
edited and improved the manuscript and figures.

## Competing interests

Authors declare no conflict of interest.

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
