# Peer review of "Exploring the possible role of satellite-based rainfall data to estimate inter- and intra-annual global rainfall erosivity"

_Hydrology and Earth System Sciences, 2021_

## Author Comment (AC1)

**Reviewer #1**

*R1C0 Comment*: The paper investigated the possible role of satellite-based rainfall data to estimate rainfall erosivity at global, continental and local scales. Besides, the application of a simple-linear function for CMORPH data correction was also conducted in this paper. The paper is interesting and is well organized. The layout of the manuscript conveys a clear presentation of the topic. However, I do have few questions regarding the content and results of this paper. Some major queries should be clarified before acceptance.

*R1C0 Response*: We would like to thank Reviewer #1 for reviewing our manuscript. We very much appreciate the encouraging comments and overall positive evaluation on our study. Point-by-point detailed responses to the specific comments are provided below. Thanks.

**General comments**

*R1C1 Comment*: Bothe abstract and conclusion should be improved. The authors should emphasize the contribution of this paper.

*R1C1 Response*: Thanks for highlighting this aspect. We see the point of Reviewer #1 and as suggested both abstract and conclusions will be modified to indicate the main contribution of this paper to the field. The main contribution is that we derived global rainfall erosivity based on the high-frequency (30-min) satellite-based product (CMORPH) and compare it to the station-based rainfall erosivity data (GloREDa). To the best of author's knowledge, all previous global studies that focused on satellite-based products used daily or monthly data and lacked the ability to compare their predictions with hourly and sub-hourly rainfall records. Therefore, such high-frequency data (CMORPH) can now be used as input to dynamic global soil erosion models with a better understanding of its performances and limitations.

*R1C2 Comment*: As the authors mentioned in the manuscript that many studies have conducted the satellite-based precipitation products for rainfall erosivity estimations. I wonder what's the difference between this paper and previous studies. Is there any significant improvement or contribution obtained in this paper?

*R1C2 Response*: Thanks for your remark. As indicated, all previous studies applied either daily or monthly data while only a few studies used high-frequency data but focused on local or regional scales and none of these at global scale. Thus, in our opinion this is the relevant difference and novelty in comparison to previous studies that used satellite-based data for the estimation of the global rainfall erosivity. Here, we wanted to test this new use of the satellite-based data and acquire a better understanding of the areas where results are adequate and those where significant under- or overestimations occur.

*R1C3 Comment*: Parts of the description are not in accord with Figures and Tables in the manuscript. For example, Fig. 1 (lines 159-160) and Table 3 (lines 213-215). Please check throughout the manuscript.

*R1C3 Response*: Noted with thanks. Yes, there are some technical issues that will be corrected in the revised version. Thank you for pointing to these issues.

*R1C4 Comment*: Table 1. The mean values calculated by CMORPH and ED indicated a significant different trend for Africa and Asia. Please provide possible reason.

*R1C4 Response*: Thanks for your remark. As suggested by the Reviewer #1 additional discussion about these results will be added to the manuscript. The main reason for differences lies in the fact that the ED concept indirectly uses the GloREDa results produced by Panagos et al. (2017). For example, for Africa only a small number of stations was used for the calculation of the global rainfall erosivity by Panagos et al. (2017). While for Asia a spatial rainfall erosivity pattern is similar (can also be seen from a similar Gini value in Table 1) but the absolute values of rainfall erosivity are different, which can be attributed to the issues related to the detection of rainfall by satellite-based products in mountainous regions (e.g., Stampoulis and Anagnostou, 2012) as well as limited amount of gauge-based data in data scare regions of Asia.

*R1C5 Comment*: Results obtained from CMORPH reveal a serious underestimation problem for annual scale, whereas results obtained for monthly scale overestimate the rainfall erosivity for six months. I wonder if this is reasonable.

*R1C5 Response*: Thanks for the comment. It should be noted that monthly rainfall erosivity comparison was only done for Europe (Table 4), since a global monthly rainfall erosivity maps have not yet been produced. Thus, Table 4 caption will be modified to better indicate this. Annual rainfall erosivity for Europe is more similar (Table 3).

*R1C6 Comment*: I am curious what is the CMORPH correction procedure? How do you get the equation (5)? It doesn't make sense to me that the correction equation did not adopt the information of CMORPH.

*R1C6 Response*: Noted with thanks. Both reviewers expressed some concerns regarding the proposed correction procedure. We see their point and we are willing to rephrase or remove this additional analysis from the revised version of the manuscript. The correction rests on a simple mathematical expression. This part can be removed since the proposed method was not very complex and it was more suitable for some engineering applications and not detailed scientific studies. Additional discussion about the needed corrections of the CMOPRH data will be added to the revised version of the manuscript. Thanks.

**Other comments**

*R1C7 Comment*: Line 187. What's R approach?

*R1C7 Response*: Thanks for your remark. R will be replaced with "rainfall erosivity" in the revised version.

*R1C8 Comment*: Line 189. Replace Oceania with North America (see Table 1).

*R1C8 Response*: Indeed, there was an error, this will be corrected. Thanks.

*R1C9 Comment*: Parts of the values displayed in Table 3 are incorrect. -40%, +11% and -56% (remove the %).

*R1C9 Response*: Noted with thanks. The "%" will be removed.

*R1C9 Comment*: Parts of the values displayed in Table 3 are incorrect. -40%, +11% and -56% (remove the %).

*R1C9 Response*: Noted with thanks. The "%" will be removed.

---

## Author Comment (AC2)

**Reviewer #2**

*R2C0 Comment*: This manuscript considers the applicability of satellite-based rainfall data to estimate global rainfall erosivity at multiple scales. The paper is intriguing and the potential for using satellite-based rainfall to achieve global data is promising. However, I have several concerns and should be considered before acceptance.

There are numerous grammatical errors throughout the manuscript. I suggest a thorough proofreading and perhaps a professional editing service. Also, as mentioned by Anonymous Referee #1, there are several errors in the text (ex. L159-160, text for second and third examples are switched compared to Fig1). Please check your manuscript thoroughly and reorganize for better comprehension.

*R2C0 Response*: We would like to thank Reviewer #2 for reviewing our manuscript. We very much appreciate the encouraging comments and overall positive evaluation on our study. Point-by-point detailed responses to the specific suggestions are provided below. Thanks.

As indicated in the response to the Reviewer #1, we will correct all highlighted smaller technical issues and typos. We will also have a throughout proofreading and correct potential issues and if needed (perhaps based on editor evaluation) we will contact editing service.

**Specific comments**

*R2C1 Comment*: L217-221: I could not understand this section, especially L216-218. Is the Gini[/] in table 3 the ratio of CMORPH gini to GloREDa gini? If so, how can we interpret this is better than bias of mean values? Please elaborate.

*R2C1 Response*: This is a good suggestion. Thanks. Additional discussions about the usage of the Gini coefficient will be included in the revised version of the manuscript. The Gini coefficient is a single number that demonstrates a degree of inequality in a distribution of income/wealth. Here, it is used to captures the inequality in the spatial distribution of rainfall erosivity. Accordingly, similar values of Gini coefficient indicate that spatial patterns are similar. Values shown in Table 3 are bias values of calculated Gini coefficients, as Reviewer #2 indicated correctly. Gini coefficient was meant to be used as an additional metric that would capture the spatial distribution of the rainfall erosivity. Thus, it should be looked together with the bias of the mean values to get a more holistic view on the differences between rainfall erosivity maps. For example, mean erosivity per continent could be similar but we could have an overestimation in one area and overestimation in other part.

*R2C2 Comment*: L231-L239: Are the pearson correlation of mean annual rainfall erosivity and gini coefficient calculated using basin averaged mean annual rainfall erosivity? Please elaborate on the calculation, especially how the spatial distribution of each sub-catchment is considered.

*R2C2 Response*: Noted with thanks. Indeed, mean annual rainfall erosivity per catchment was calculated. Additional description will be included in the revised version as suggested by the Reviewer #2.

*R2C3 Comment*: L301-L314: I could not understand how equation 5 is derived and applied. Please clarify.

*R2C3 Response*: Thanks for your remark. As already discusses with Reviewer #1, we are willing to remove this part from the revised version of the manuscript. Additional discussion about the needed corrections of the CMOPRH data will be added to the revised version of the manuscript. Thanks.

*R2C4 Comment*: L327-L328: How can this be said from the limited amount of grids with a significant trend?

*R2C4 Response*: Noted with thanks. As suggested by the Reviewer #2 this sentence will be modified or removed in the revised version.

*R2C5 Comment*: L335-L339: In table 3, CMORPH in North America is largely underestimated, whereas Kim et al (2020) reports CMORPH in US in overestimated. If CMORPH in this study is compared for only US, does it show an overestimation similar to Kim et al (2020)? If not, please elaborate on the difference.

*R2C5 Response*: Thanks for your remark. Please note that Kim et al. (2020) wrote (section 3.3, please also see Figure 9 in Kim et al., 2020): "*The range of the R-factors in Panagos et al. (2017) is 6–9645 MJ mm $ha^{-1}$ $h^{-1}$ $yr^{-1}$, and the mean value is 2067 MJ mm $ha^{-1}$ $h^{-1}$ $y^{r-1}$, i.e., 1.65 times higher than the mean R-factor estimated in this study*". Thus, values obtained according to the CMORPH were smaller compared to the GloREDa (Panagos et al., 2017). This is consistent to what is shown in our Table 3. Our results are in agreement to what was reported by Kim et al. (2020). Please also note that Kim et al. (2020) indicated an overestimation of rainfall erosivity near water bodies and this kind of overestimation was also detected in our study.

*R2C6 Comment*: L343-L361: Information on CMORPH precipitation accuracy in different regions does not seem relevant unless it is clear to readers how it affects the over/underestimations of CMORPH rainfall erosivity in those regions.

*R2C6 Response*: Good point, thanks. As suggested by the Reviewer #2 these sentences will modified in order to indicate a link between precipitation and rainfall erosivity.

**Minor comments**

*R2C7 Comment*: L11-12: I could not understand what "As this data scarcity is likely to characterize the upcoming years" means.

*R2C7 Response*: Thanks for your remark. It was meant that since the density of gauge-based data will most likely not increase in future, alternative data sources could be useful. However, the sentence will be modified in order to make it more clear for the readers.

*R2C8 Comment*: L198: This is not a sentence.

*R2C8 Response*: Noted with thanks. This sentence will be modified.

*R2C9 Comment*: L202: the comparison of 1981-2019 does not seem relevant for this manuscript.

*R2C9 Response*: Indeed, most of the investigations were performed using data after 1998. Thus, this sentence will be modified.

*R2C10 Comment*: L220: CMORPH seems to be better for Europe? Please clarify.

*R2C10 Response*: Indeed, the sentence will be modified.

*R2C11 Comment*: L267-268: How can this be said?

*R2C11 Response*: Noted with thanks. This sentence will be modified in the revised version.

*R2C12 Comment*: Figure 6: There are no dotted lines.

*R2C12 Response*: Indeed, figure caption will be corrected.

*R2C13 Comment*: Figure 9: What is the blue dotted line?

*R2C13 Response*: Thanks for your remark. The blue dotted line is a linear trend line of the original CMORPH data. However, please note that this section will be removed in the revised version.

---

## Author Response (AR1)

Ljubljana, 21th of January 2022

**TO:**
**Editorial Office**
**Hydrology and Earth System Sciences**

Dear Editorial Office,

Please find enclosed the revised version of the original manuscript entitled "*Exploring the possible role of satellite-based rainfall data to estimate inter- and intra-annual global rainfall erosivity*" authored by N. Bezak, P. Borrelli and P. Panagos.

We would like to thank the editor and the reviewers for their time and efforts in reviewing our manuscript, for the constructive comments and observations that can help us to bring the manuscript up to HESS standard, and for the opportunity to address the highlighted concerns and suggestions in this resubmission.

In response to the reviewers' comments, we have conducted a major revision and incorporated new material in the revised version. Most notably: (1) some parts of the manuscript that were highlighted as potentially less relevant were removed, (2) additional discussion was added as suggested by both reviewers, (3) elements of novelty were highlighted, and (4) several smaller corrections that were suggested by two reviewers and associate editor were incorporated into the manuscript.

Please find below a detailed response to all of the reviewer's comments. All changes are indicated with red coloured text in the manuscript.

Sincere regards on behalf of all the authors,

Nejc Bezak

**Editor**

*EC0 Comment*: Dear authors,

Comments from two reviewers have returned. Both reviewers give seriours concerns about the contributions of your manuscript and numerous grammatical errors. Therefore, I hope the authors can ask a professional editing service for help and revise the manuscript carefully based on two reviewers' comments. I have some additional comments.

*EC0 Response*: We would like to thank Editor for reviewing our manuscript. We very much appreciate the encouraging comments and overall positive evaluation on our study. We did contact a native English speaker to help with the editing in order to improve the grammar, overall style and English spelling. These changes are not shown with colored text in the revised version in order obtain the visibility of other modifications related to the content of the paper and reviewers comments. Other parts of the manuscript were corrected as well. Point-by-point detailed responses to the specific comments are provided below. Thanks.

*EC1 Comment*: 1) According to the authors, the ED was obtained by using the global rainfall erosivity map by Panagos et al. (2017). However, the GloREDa point dataset was again used to evaluate the performance of the rainfall erosivity derived using the ED concept. Does this make sense?

*EC1 Response*: Thanks for your observation. In order to reply to your question, and to report this aspect more clearly in the manuscript, the relevant part was modified in the revised version of the manuscript. It should be underlined that ED is computed from rainfall erosivity (Panagos et al., 2017), witch, in turn, is developed using as input (i) GloREDa storms rainfall erosivity and (ii) an independent set of bioclimatic variable (source = WorldClim) for their spatial interpolation. Here, ED is then computed with a third independent rainfall dataset, i.e. ERA5. We better describe these aspects in the new text. We recognize that this is not yet the best case scenario but since there is no other global rainfall erosivity map developed that would be constructed using high-temporal resolution data, we could only use the GloREDa data as input for the ED. We now emphasized this aspects (pros and cons) in several parts of the manuscript including the Conclusions. Thank you.

*EC2 Comment*: 2) I don't suggest the removal of the correction part. I think this part is rather useful. The authors may explain how equation 5 was derived and applied, rather than just removing this part.

*EC2 Response*: Thank you for highlighting this aspect. Please note that the part under discussion was maintained in the revised version of the manuscript. It was further expanded and improved based on results obtained through new regional statistical analysis (more details in the revised section 3.4). Additional discussion about the topic was added as well. The decision to provide new regional correction factors per climate zones rests on the goal to provide more meaningful correction compared to just one correction factor at global scale.

*EC3 Comment*: 3) In Page 6, 'second' should be 'third' and 'third' should be 'second', right?

*EC3 Response*: Indeed, you are right. Thanks for highlighting this matter which was fully addressed in the revised version.

**Reviewer #1**

*R1C0 Comment*: The paper investigated the possible role of satellite-based rainfall data to estimate rainfall erosivity at global, continental and local scales. Besides, the application of a simple-linear function for CMORPH data correction was also conducted in this paper. The paper is interesting and is well organized. The layout of the manuscript conveys a clear presentation of the topic. However, I do have few questions regarding the content and results of this paper. Some major queries should be clarified before acceptance.

*R1C0 Response*: We would like to thank Reviewer #1 for reviewing our manuscript. We very much appreciated the encouraging comments and overall positive evaluation on our study. Point-by-point detailed responses to the specific comments are provided below. Thanks.

**General comments**

*R1C1 Comment*: Bothe abstract and conclusion should be improved. The authors should emphasize the contribution of this paper.

*R1C1 Response*: Thanks for highlighting this aspect. We see the point of Reviewer #1 and as suggested both abstract and conclusions were modified to further highlight the main contribution of this paper to the field. A highly relevant contribution rests on the computation of global rainfall erosivity data based on the high-frequency (30-min) satellite-based product (CMORPH) and the subsequent comparison with hourly and sub-hourly station-based rainfall erosivity records (GloREDa). To the best of author's knowledge, all previous global studies that focused on satellite-based products used daily or monthly data. In addition, they all lacked the ability to compare their predictions with hourly and sub-hourly rainfall records and over a large number of geoclimatic zones. We believe that the gained insights can help our community to move towards a better understanding of such high-frequency data (CMORPH), and how it can be used as input for global soil erosion models with a more profound understanding of its performances and limitations.

*R1C2 Comment*: As the authors mentioned in the manuscript that many studies have conducted the satellite-based precipitation products for rainfall erosivity estimations. I wonder what's the difference between this paper and previous studies. Is there any significant improvement or contribution obtained in this paper?

*R1C2 Response*: Thanks for your remark. As indicated, the vast majority of previous studies applied either daily or monthly data. Only a few studies used high-frequency data but these still maintained their applications at local or regional scales. We present for the first time a global scale application. An approach capable to observe how these data sets perform over different geographic and climatic regions. This, in our opinion makes a relevant difference compared to previous applications providing novelty. Here, we have more extensively tested the use of this data gaining a broader understanding of the areas where results are adequate and those where significant under- or overestimations tend to occur. Please note that the manuscript has been expertly adjusted to make the novelty aspects clearer for the reader. Thanks.

*R1C3 Comment*: Parts of the description are not in accord with Figures and Tables in the manuscript. For example, Fig. 1 (lines 159-160) and Table 3 (lines 213-215). Please check throughout the manuscript.

*R1C3 Response*: Noted with thanks. Yes, there are some technical issues that were corrected in the revised version. Thank you for pointing to these issues.

*R1C4 Comment*: Table 1. The mean values calculated by CMORPH and ED indicated a significant different trend for Africa and Asia. Please provide possible reason.

*R1C4 Response*: Thanks for your remark. As suggested by the Reviewer #1 additional discussion about these results were added to the manuscript. The main reason for differences lies in the fact that the ED concept indirectly uses the GloREDa results produced by Panagos et al. (2017). For example, for Africa only a small number of stations was used for the calculation of the global rainfall erosivity by Panagos et al. (2017). While for Asia a spatial rainfall erosivity pattern is similar (can also be seen from a similar Gini value in Table 1) but the absolute values of rainfall erosivity are different, which can be attributed to the issues related to the detection of rainfall by satellite-based products in mountainous regions (e.g., Stampoulis and Anagnostou, 2012) as well as limited amount of gauge-based data in data scare regions of Asia.

*R1C5 Comment*: Results obtained from CMORPH reveal a serious underestimation problem for annual scale, whereas results obtained for monthly scale overestimate the rainfall erosivity for six months. I wonder if this is reasonable.

*R1C5 Response*: Thanks for the comment. It should be noted that monthly rainfall erosivity comparison was only done for Europe (Table 4), since monthly global rainfall erosivity maps are not yet produced. Thus, Table 4 caption was modified to better indicate this. Annual rainfall erosivity for Europe is more similar (Table 3). Underestimation (CMORPH compared to GloREDa) in summer for Europe could be attributed to the detection issues of the most extreme summer thunderstorm while bias in winter is high but absolute rainfall erosivity values in this part of year are relatively low in most of the Europe.

*R1C6 Comment*: I am curious what is the CMORPH correction procedure? How do you get the equation (5)? It doesn't make sense to me that the correction equation did not adopt the information of CMORPH.

*R1C6 Response*: Noted with thanks. Please kindly note that this part of the methodology was further elaborated and analyzed to provide a better description of the methods and goals. As a matter of fact, in the revised version we now apply multiple correction factors divided per climate zones. Relevant parts in the Results, Discussion and Conclusions sections were also updated accordingly. Thanks.

**Other comments**

*R1C7 Comment*: Line 187. What's R approach?

*R1C7 Response*: Thanks for your remark. R was be replaced with "rainfall erosivity" in the revised version.

*R1C8 Comment*: Line 189. Replace Oceania with North America (see Table 1).

*R1C8 Response*: Indeed, there was an error, this was corrected. Thanks.

*R1C9 Comment*: Parts of the values displayed in Table 3 are incorrect. -40%, +11% and -56% (remove the %).

*R1C9 Response*: Noted with thanks. The "%" was be removed.

**Reviewer #2**

*R2C0 Comment*: This manuscript considers the applicability of satellite-based rainfall data to estimate global rainfall erosivity at multiple scales. The paper is intriguing and the potential for using satellite-based rainfall to achieve global data is promising. However, I have several concerns and should be considered before acceptance.

There are numerous grammatical errors throughout the manuscript. I suggest a thorough proofreading and perhaps a professional editing service. Also, as mentioned by Anonymous Referee #1, there are several errors in the text (ex. L159-160, text for second and third examples are switched compared to Fig1). Please check your manuscript thoroughly and reorganize for better comprehension.

*R2C0 Response*: We would like to thank Reviewer #2 for reviewing our manuscript. We very much appreciated the encouraging comments and overall positive evaluation on our study. Point-by-point detailed responses to the specific suggestions are provided below. Thanks.

As indicated in the response to the Reviewer #1, all highlighted smaller technical issues and typos were corrected. A throughout proofreading and correction of potential issues was done with help of the native speaker.

**Specific comments**

*R2C1 Comment*: L217-221: I could not understand this section, especially L216-218. Is the Gini[/] in table 3 the ratio of CMORPH gini to GloREDa gini? If so, how can we interpret this is better than bias of mean values? Please elaborate.

*R2C1 Response*: This is a good suggestion. Thanks. Additional discussions about the usage of the Gini coefficient was included in the revised version of the manuscript. The Gini coefficient is a single number that demonstrates a degree of inequality in a distribution of income/wealth. Here, it is used to captures the inequality in the spatial distribution of rainfall erosivity. Accordingly, similar values of Gini coefficient indicate that spatial patterns are similar. Values shown in Table 3 are bias values of calculated Gini coefficients, as Reviewer #2 indicated correctly. Gini coefficient was meant to be used as an additional metric that would capture the spatial distribution of the rainfall erosivity. Thus, it should be looked together with the bias of the mean values to get a more holistic view on the differences between rainfall erosivity maps. For example, mean erosivity per continent could be similar but we could have an overestimation in one area and overestimation in other part.

*R2C2 Comment*: L231-L239: Are the pearson correlation of mean annual rainfall erosivity and gini coefficient calculated using basin averaged mean annual rainfall erosivity? Please elaborate on the calculation, especially how the spatial distribution of each sub-catchment is considered.

*R2C2 Response*: Noted with thanks. Indeed, mean annual rainfall erosivity per catchment was calculated. Additional description was added to the section 3.3.2 as suggested by the Reviewer #2.

*R2C3 Comment*: L301-L314: I could not understand how equation 5 is derived and applied. Please clarify.

*R2C3 Response*: Thanks for your remark. Please kindly note that this part of the methodology was further elaborated and analyzed to provide a better description of the methods and goals. As a matter of fact, in the revised version we now apply multiple correction factors divided per climate zones. Relevant parts in the Results, Discussion and Conclusions sections were also updated accordingly. Thanks.

*R2C4 Comment*: L327-L328: How can this be said from the limited amount of grids with a significant trend?

*R2C4 Response*: Noted with thanks. As suggested by the Reviewer #2 this sentence was removed in the revised version.

*R2C5 Comment*: L335-L339: In table 3, CMORPH in North America is largely underestimated, whereas Kim et al (2020) reports CMORPH in US in overestimated. If CMORPH in this study is compared for only US, does it show an overestimation similar to Kim et al (2020)? If not, please elaborate on the difference.

*R2C5 Response*: Thanks for your remark. Please note that Kim et al. (2020) wrote (section 3.3, please also see Figure 9 in Kim et al., 2020): "*The range of the R-factors in Panagos et al. (2017) is 6–9645 MJ mm ha$^{-1}$ h$^{-1}$ yr$^{-1}$, and the mean value is 2067 MJ mm ha$^{-1}$ h$^{-1}$ y$^{r-1}$, i.e., 1.65 times higher than the mean R-factor estimated in this study*". Thus, values obtained according to the CMORPH were smaller compared to the GloREDa (Panagos et al., 2017). This is consistent to what is shown in our Table 3. Our results are in agreement to what was reported by Kim et al. (2020). Please also note that Kim et al. (2020) indicated was overestimation of rainfall erosivity near water bodies and this kind of overestimation was also detected in our study.

*R2C6 Comment*: L343-L361: Information on CMORPH precipitation accuracy in different regions does not seem relevant unless it is clear to readers how it affects the over/underestimations of CMORPH rainfall erosivity in those regions.

*R2C6 Response*: Good point, thanks. As suggested by the Reviewer #2 these sentences were modified in order to indicate a link between precipitation and rainfall erosivity.

**Minor comments**

*R2C7 Comment*: L11-12: I could not understand what "As this data scarcity is likely to characterize the upcoming years" means.

*R2C7 Response*: Thanks for your remark. It was meant that since the density of gauge-based data will most likely not increase in future, alternative data sources could be useful. However, the sentence was modified in order to make it more clear for the readers.

*R2C8 Comment*: L198: This is not a sentence.

*R2C8 Response*: Noted with thanks. This sentence was be removed.

*R2C9 Comment*: L202: the comparison of 1981-2019 does not seem relevant for this manuscript.

*R2C9 Response*: Indeed, most of the investigations were performed using data after 1998. Thus, this sentence was modified.

*R2C10 Comment*: L220: CMORPH seems to be better for Europe? Please clarify.

*R2C10 Response*: Indeed, the sentence was be modified.

*R2C11 Comment*: L267-268: How can this be said?

*R2C11 Response*: Noted with thanks. This sentence was removed in the revised version.

*R2C12 Comment*: Figure 6: There are no dotted lines.

*R2C12 Response*: Indeed, figure caption was corrected.

*R2C13 Comment*: Figure 9: What is the blue dotted line?

*R2C13 Response*: Thanks for your remark. Figure 9 was replaced with new figure since new correction factors (per climate zone) were used.

---

## Editor Decision (ED1)

Hydrol. Earth Syst. Sci. Discuss., referee comment RC1
https://doi.org/10.5194/hess-2021-417-RC1, 2021
**Comment on hess-2021-417**

Anonymous Referee #1

Referee comment on "Exploring the possible role of satellite-based rainfall data to estimate inter☐ and intra☐annual global rainfall erosivity" by Nejc Bezak et al., Hydrol. Earth Syst. Sci. Discuss., https://doi.org/10.5194/hess-2021-417-RC1, 2021

The paper investigated the possible role of satellite-based rainfall data to estimate rainfall erosivity at global, continental and local scales. Besides, the application of a simple-linear function for CMORPH data correction was also conducted in this paper. The paper is interesting and is well organized. The layout of the manuscript conveys a clear presentation of the topic. However, I do have few questions regarding the content and results of this paper. Some major queries should be clarified before acceptance.

General comments:

- Bothe abstract and conclusion should be improved. The authors should emphasize the contribution of this paper.
- As the authors mentioned in the manuscript that many studies have conducted the satellite-based precipitation products for rainfall erosivity estimations. I wonder what's the difference between this paper and previous studies. Is there any significant improvement or contribution obtained in this paper?
- Parts of the description are not in accord with Figures and Tables in the manuscript. For example, Fig. 1 (lines 159-160) and Table 3 (lines 213-215). Please check throughout the manuscript.
- Table 1. The mean values calculated by CMORPH and ED indicated a significant different trend for Africa and Asia. Please provide possible reason.
- Results obtained from CMORPH reveal a serious underestimation problem for annual scale, whereas results obtained for monthly scale overestimate the rainfall erosivity for six months. I wonder if this is reasonable.
- I am curious what is the CMORPH correction procedure? How do you get the equation (5)? It doesn't make sense to me that the correction equation did not adopt the information of CMORPH.

Other comments:

- Line 187. What's R approach?
- Line 189. Replace Oceania with North America (see Table 1).
- Parts of the values displayed in Table 3 are incorrect. -40%, +11% and -56% (remove the %).

[Figure]

Hydrol. Earth Syst. Sci. Discuss., referee comment RC2
https://doi.org/10.5194/hess-2021-417-RC2, 2021
**Comment on hess-2021-417**

Anonymous Referee #2
* * *
Referee comment on "Exploring the possible role of satellite-based rainfall data to estimate inter☐ and intra☐annual global rainfall erosivity" by Nejc Bezak et al., Hydrol. Earth Syst. Sci. Discuss., https://doi.org/10.5194/hess-2021-417-RC2, 2021
* * *
This manuscript considers the applicability of satellite-based rainfall data to estimate global rainfall erosivity at multiple scales. The paper is intriguing and the potential for using satellite-based rainfall to achieve global data is promising. However, I have several concerns and should be considered before acceptance.

**General comment:**

There are numerous grammatical errors throughout the manuscript. I suggest a thorough proofreading and perhaps a professional editing service. Also, as mentioned by Anonymous Referee #1, there are several errors in the text (ex. L159-160, text for second and third examples are switched compared to Fig1). Please check your manuscript thoroughly and reorganize for better comprehension.

**Specific comments:**

- L217-221: I could not understand this section, especially L216-218. Is the Gini[/] in table 3 the ratio of CMORPH gini to GloREDa gini? If so, how can we interpret this is better than bias of mean values? Please elaborate.
- L231-L239: Are the pearson correlation of mean annual rainfall erosivity and gini coefficient calculated using basin averaged mean annual rainfall erosivity? Please elaborate on the calculation, especially how the spatial distribution of each sub-catchment is considered.
- L301-L314: I could not understand how equation 5 is derived and applied. Please clarify.
- L327-L328: How can this be said from the limited amount of grids with a significant trend?
- L335-L339: In table 3, CMORPH in North America is largely underestimated, whereas Kim et al (2020) reports CMORPH in US in overestimated. If CMORPH in this study is compared for only US, does it show an overestimation similar to Kim et al (2020)? If not, please elaborate on the difference.
- L343-L361: Information on CMORPH precipitation accuracy in different regions does not seem relevant unless it is clear to readers how it affects the over/underestimations of CMORPH rainfall erosivity in those regions.

**Minor comments:**

- L11-12: I could not understand what "As this data scarcity is likely to characterize the upcoming years" means.
- L198: This is not a sentence.
- L202: the comparison of 1981-2019 does not seem relevant for this manuscript.
- L220: CMORPH seems to be better for Europe? Please clarify.
- L267-268: How can this be said?
- Figure6: There are no dotted lines.
- Figure9: What is the blue dotted line?